# Obtention and Characterization of Ferrous Chloride FeCl_2_·4H_2_O from Water Pickling Liquors

**DOI:** 10.3390/ma14174840

**Published:** 2021-08-26

**Authors:** Lorena Alcaraz, Belén Sotillo, José F. Marco, Francisco J. Alguacil, Paloma Fernández, Félix A. López

**Affiliations:** 1Centro Nacional de Investigaciones Metalúrgicas (CENIM), Consejo Superior de Investigaciones Científicas (CSIC), Avda. Gregorio del Amo, 8, 28040 Madrid, Spain; fjalgua@cenim.csic.es (F.J.A.); f.lopez@csic.es (F.A.L.); 2Departamento de Física de Materiales, Facultad de Ciencias Físicas, Universidad Complutense de Madrid (UCM), 28040 Madrid, Spain; bsotillo@fis.ucm.es (B.S.); arana@fis.ucm.es (P.F.); 3Instituto de Química Física ‘Rocasolano’, Consejo Superior de Investigaciones Científicas (CSIC), C/Serrano, 119, 28006 Madrid, Spain; jfmarco@iqfr.csic.es

**Keywords:** ferrous chloride, FeCl_2_·4H_2_O crystals, water pickling liquors

## Abstract

As a hazardous waste, water pickling liquors must be properly treated. An alternative consists of promoting the formation of ferrous salts from this residue due to their higher ferrous content. Since FeCl_2_·4H_2_O is widely used in several applications, obtaining pure crystals of this material appears to be an interesting prospect. However, this compound has scarcely been investigated. In the present work, FeCl_2_·4H_2_O crystals were obtained from water pickling liquors. Their structural and morphological characteristics were investigated by X-ray diffraction, scanning electron microscopy as well as Mössbauer spectroscopy. In addition, the photoluminescence study of the obtained samples was also assessed. It was observed that after some aging time, the obtained crystals changed in colour from green to more yellowish. As such, the aged sample was also evaluated, and their structural characteristics were compared with the original crystals. Despite this, the obtained crystals exhibit a FeCl_2_·4H_2_O structure, which is not modified with the aging of the sample.

## 1. Introduction

Water pickling liquors is a common waste generated in the process of steel pickling and the electroplating industry and usually contain an acid (cleaning reagent) and a large amount of iron ions in their composition [1,2]. Normally, cleaning agents such as hydrochloric, sulphuric, hydrofluoric, and nitric acids are widely used in the mentioned process [1]. Due to the acidic nature (30–100 g/L concentration of acid), and the high concentration of iron ions (at around 60–250 g/L), this type of waste can lead to severe environmental damage, and it has been listed as hazardous waste [2]. For this reason, an adequate treatment for water pickling liquors is of great importance [3,4].

Several techniques have been used to treat water pickling liquors including electrodialysis [5], evaporation [6], crystallization [7], selective precipitation [8], neutralization [5], or ion exchange [9]. In addition, previous investigations have reported the obtention of functional materials from water pickling liquors such as magnetic biochar [10], composites inorganic coagulants [11], or ferrites [12,13].

Another alternative is the direct conversion of the unused acid to its corresponding iron salt [14]. Among the different ferrous salts, there is a great interest in FeX_2_·4H_2_O systems, especially FeCl_2_·4H_2_O, due to their potential applications in chemistry and composite material areas [15]. FeCl_2_·4H_2_O is commonly used as a reducing [16,17] and electrolytic agent [18], for the synthesis of pharmaceutical compounds [19,20], or for water treatment [21,22].

Despite ferrous chloride being obtained from water pickling liquors, to the best of our knowledge, the structural characterization of FeCl_4_·4H_2_O has been scarcely investigated. In the present work, the obtention of FeCl_2_·4H_2_O from water pickling liquors using an effective process is described. A deep characterization of the structural and optical properties of the obtained samples was performed. In addition, the effects of aging on the crystal structure, structural characteristics as well as optical properties have been conducted.

## 2. Experimental

### 2.1. Obtention of the Iron Chloride from Water Pickling Liquors

Ferrous chloride was obtained from the water pickling liquors from the hydrochloric acid (HCl) pickling process of carbon steel. When carbon steel exposed to air is treated at temperatures between the range of 575–1370 °C, it leads to an oxide layer usually forming on the surface. This oxide results in scale formation and consists of three well-defined layers, the thickness and composition of which depend on the time and temperature of the treatment. The scale tends to protect the metal from further oxidation at high temperatures. Generally, this scale is formed by a thick layer of wustite (with FeO formula), a magnetite layer (Fe_3_O_4_), and a thin layer of hematite (Fe_2_O_3_). The removal of the scale is conducted by steel pickling using a hydrochloric acid solution. Subsequently, the steel is washed with water to eliminate the remaining acid and the dissolved iron. Then, the superficial oxides are dissolved, which leads to water pickling liquors.

The oxidation layer composed of different types of iron oxides (Fe_2_O_3_, Fe_3_O_4_, and FeO) reacts with HCl to form ferrous chloride through reactions 1–3 [23]. Meanwhile, hydrochloric acid penetrates the oxidation layer and attacks the underlying metal. Ferrous chloride and hydrogen gas are produced by the reaction between hydrochloric acid and base steel (reaction 4) [23]. Due to these reactions, the oxidized layer is peeled off of the surface, which is the main action of the pickling process.
(1)$${\mathrm{Fe}}_{2}O_{3}+\mathrm{Fe}+6\mathrm{HCl}\;\to \;3{\mathrm{FeCl}}_{2}+3H_{2}O$$
(2)$${\mathrm{Fe}}_{3}O_{4}+\mathrm{Fe}+8\mathrm{HCl}\;\to \;4{\mathrm{FeCl}}_{2}+4H_{2}O$$
(3)$$\mathrm{FeO}+2\mathrm{HCl}\;\to {\;\mathrm{FeCl}}_{2}+H_{2}O$$
(4)$$\mathrm{Fe}+2\mathrm{HCl}\;\to {\;\mathrm{FeCl}}_{2}+H_{2}$$

When free HCl is exhausted and ferrous chloride accumulates in the water pickling liquors to the point where pickling cannot be conducted effectively, spent pickling solutions are discharged from the pickling tanks and are managed as hazardous waste. It is estimated that around 380,000 m^3^ of chlorinated pickling water is produced annually in Europe [24]. One way to utilize this water is to treat it to recover the ferrous chloride and residual acid.

The water pickling liquors in this work come from the company HIASA (Asturias, Spain). The acid content was determined by titration with standard NaOH. For this titration, NaOH 0.1 N was slowly added to a mixture of water pickling liquor and distilled water (1:50, *v*/*v*) until pH values of 2 (free acid) and 9 (total acid) were achieved. The liquor (139.9 ± 5.1 g/L Fe^2+^, 43.6 ± 8.2 g/L free acid, 237.5 ± 20.5 g/L total acid) was treated in a pilot evaporation-crystallization plant.

The pickling water was heated to 130 °C in a pre-heater before entering a 500 mm long Vigreux column at the same temperature. The pickling water inlet flowed into the column was 3 L/h. In the column, the pickling water was split into two streams: a vapour stream that condenses in a condenser at 65 °C and a liquor enriched in Fe^2+^ (188 ± 8.6 g/L). The vapour leaving the Vigreux column was split into two streams in the condenser. A concentrated acid stream (107.7 ± 21.8 g/L) and a water stream, which was treated in a condenser at 6 °C to obtain a water stream without free acid. The whole system was kept under a vacuum of −640 mbar. The enriched liquor was crystallized in a crystallizer at 130 °C and was filtered to obtain a mass of crystallized ferrous chloride and a residual solution that was recirculated at the beginning of the process. The final obtained crystals were washed twice in order to eliminate the acid that could potentially remain on the surface of the crystals. Finally, the crystals were vacuum dried at 60 °C for 12 h. Figure 1 shows the ferrous chloride crystals immediately after crystallization and filtration. The concentrated acid and water could be reused in the pickling process.

In order to analyze the effect of aging in the samples, the obtained crystals were aged for different times. Table 1 the exhibits nomenclature of the samples assessed in the present work as well as the description of the different aging conditions are also shown in it.

### 2.2. Characterization of the Iron Chloride Obtained from Water Pickling Liquors

X-ray diffraction measurements (XRD) have been conducted using a PANalytical Empyrean diffractometer (PANalytical, Almelo, The Netherlands)) using λ Cu Kα radiation with a step size of 0.02° (2θ) and a step time of 7.9 s. Powder diffraction data were refined using the Rietveld method using the FullProf software [25].

Micro-Raman measurements were performed at room temperature in a Horiba Jobin Yvon LabRAM HR800 (Horiba JobinYvon, Villeneuve d’Ascq, France) confocal microscope. Raman spectra were recorded under excitation with the 632.8 nm line of a He-Ne laser (Melles Griot, NY, USA). A charge-coupled device (CCD) detector (Tokyo, Japan) was used to collect the scattered light dispersed by a 600 lines/mm grating. The spectral resolution of the system used was 1.5 cm^−1^. Photoluminescence measurements were conducted in the same system using the 325 nm line of a He-Cd laser as excitation source.

Optical micrographs were taken using a Leica DFC295 digital camera mounted on a Leica MSV266 microscope (Leica Microsystem, Wetzlar, Germany).

The morphology and chemical composition of the obtained crystals were conducted by field emission scanning electronic microscopy using a Hitachi S-4800 (Chiyoda, Japan) and a FEI Inspect SEM.

^57^Fe Mössbauer data were recorded at 298 K in the transmission mode using a conventional constant acceleration spectrometer and a ^57^Co(Rh) source. An effective absorber thickness of approximately 5 mg·cm^−2^ of natural iron was used in all of the cases. The velocity scale was calibrated using a 6 µm thick iron foil. The isomer shifts were referred to the centroid of the spectrum of α-Fe at room temperature. All the spectra were computer-fitted using Lorentzian lines.

## 3. Results and Discussion

Commercial ferrous chloride (CFC) was analyzed for comparison purposes (Figure 2a). The fresh sample (FFC) was aged for a few days (AFC) at standard conditions (around 298 K and 100 kPa of pressure). The initial light-green colour of the sample (Figure 2b) turned to brown in some areas of the obtained crystals (Figure 2c,d). The change in colour from green to more yellowish crystals occurred over the course of a few days (Figure 2c). If the samples were left in ambient atmosphere for more than 20 days, and the colour progressed towards an orange-brown (Figure 2b–d). No further changes were observed in the samples that were aged for months, indicating that at some point, the crystal degradation stops. A close inspection of the samples shows that orange regions and orange agglomerates appear between and on the crystals, respectively (Figure 2d). Moreover, although the crystals were still green in colour, they lost their transparency, and the surface was much rougher. These two effects are behind the macroscopic colour change observed in the aged samples.

### 3.1. X-ray Diffraction (XRD)

Figure 3 exhibits the comparative X-ray diffraction patterns of the samples investigated. All diffraction maxima can be indexed to a monoclinic structure with a space group P2_1_/c and Z = 2 (14) (PDF 01-071-0668) compatible with a FeCl_2_·4H_2_O phase. No additional peaks were observed within the sensitivity of the experimental that was system used, indicating the purity of the obtained samples. Lower intensity diffraction maxima can be observed in the difractogram of the CFC sample, compared to that of the FFC sample. This result could indicate the presence of an amorphous phase/thin layer of degradation products formed on the surface of the aged sample, which goes undetected by XRD. The possible formation of this thin layer on the surface of ferrous chloride crystals would lead to a decrease of the diffraction maxima intensity. It should be noted that optical images exhibit a colour change in some areas of the FFC sample (see Figure 2c,d).

In order to conduct a more detailed structural characterization, XRD data were refined by the Rietveld method, where iron atoms were distributed in the 2a (0, 0, 0) positions, and chlorine, oxygen, and hydrogen were located in the 4e (x, y, z) sites. FulProf software was used to calculate the structural parameters of the samples according to the crystallographic database code ICSD-9198 [26]. The lattice parameters and cell angles of the obtained samples are shown in Table 2. Refined R-factors obtained in these refinements are also summarized in the table. The observed, calculated, and difference XRD patterns obtained in the refinements of the FFC and AFC samples are exhibited in Figure 4. Slight differences were found in the calculated cell parameters (i.e., cell parameters and cell angles) for both the FFC and AFC ferrous chloride samples. In addition, these values were similar to previously reported values as well as to the theoretical values. The obtained results show that the structure of the FeCl_2_·4H_2_O samples obtained from the water pickling liquors is in good accordance with the theoretical structure previously reported for this compound [26,27].

According to the calculated structural parameters, the crystal structure of the obtained FeCl_2_·4H_2_O crystals structure is formed by a net of octahedra. Each octahedron has the metal (Fe) at the center surrounded by two Cl atoms and four water molecules (see Figure 5).

### 3.2. Scanning Electron Microscopy (SEM)

Figure 6 shows the SEM images of the CFC reference sample and the obtained crystals of FeCl_2_·4H_2_O (FFC fresh sample and AFC aged sample). In all cases, similar morphologies were found. SEM micrographs reveal the formation of cubic crystals, which is in good agreement with those previously reported for samples with the same stoichiometry [27]. However, a smoother surface was found in the case of the FFC sample (Figure 6b) compared to the AFC sample (Figure 6c). In the latter case, some areas of the crystals exhibited a rougher surface (shown in the inset of Figure 6c), which correspond to the orange areas of the aged sample, which might be consistent with the formation of small crystallites of a new phase on the surface of the ferrous chloride crystals. These results, again, are in agreement with the optical images and XRD results.

### 3.3. µ-Raman Spectroscopy and µ-Photoluminescence

In order to confirm the crystal structure of the obtained FeCl_2_·4H_2_O sample and to study the formation of new phases in the aging process, micro-Raman spectroscopy measurements were performed. In Figure 7, the Raman spectra of the reference CFC crystals and the fresh sample FFC are shown. In both cases (Figure 7a), all the modes can be ascribed to FeCl_2_·4H_2_O with a monoclinic C_2h_^5^ space group [28], which is in agreement with the XRD results. The 30 atoms in the unit cell give rise to 42 Raman active modes. Furthermore, the observed Raman modes can be divided into different types that are also related to different frequency ranges. At frequencies below 100 cm*^−^*^1^, octahedra external modes are detected (peak at 70 cm*^−^*^1^). The octahedra internal modes appear in the frequency range between 100 cm*^−^*^1^ and 400 cm*^−^*^1^ (Figure 7b, indicated as octahedra internal deformations, OID). The peaks at 104, 145, and 170 cm*^−^*^1^ (measured on the FFC spectrum) are related to the internal deformation bending of the octahedra. A shoulder at around 190 cm*^−^*^1^ is associated with Fe-Cl stretching modes. On the other hand, the Fe-O stretching mode peaks are located at 203 and 303 cm*^−^*^1^. The rest of the observed modes come from the vibrations of the water molecules. The band related to water vibration modes is found at 619 cm*^−^*^1^ (Figure 7a). The peaks shown in Figure 6b are ascribed to the bending vibrations of the water molecules (1632 and 1651 cm*^−^*^1^). Finally, the stretching vibrations (symmetric and antisymmetric) of the water molecules are included in the band centered at 3410 cm*^−^*^1^ (Figure 7d).

As it has been already mentioned, as the colour of the samples changes from green to orange-brown, the aspect of the crystals in the optical microscope images changes (see Figure 2c,d). There are two different regions that are observed in the AFC crystals: the region that still has a green colour and the orange region (Figure 8a). We have used micro-Raman measurements to study the differences between the green and the orange regions. The spectra recorded on the green part of the crystal (Figure 8b,c) reveal that it is still FeCl_2_·4H_2_O, as it exhibits the same modes as those described in Figure 7. The main difference is the signal intensity, which is lower in the aged samples. This can be related to a decrease in the crystal quality.

On the other hand, the spectra recorded on the orange region (Figure 8d,e) are completely different. The first observation that can be made is the disappearance of the water molecule vibrations between 1600 and 3800 cm^−1^ (Figure 8c). Then, the modes are mainly found in the range between 50 and 1000 cm^−1^ (Figure 8d). Although in some rare cases a peak at 191 cm^−1^, which is related to the FeCl_3_ compound [29], can be detected (Figure 9a), the most common peaks recorded in the orange region are those shown in Figure 8e. They are broad bands centered at about 112, 305, 399, 493, 540, and 717 cm^−1^. They can be related to different phases of iron oxyhydroxides (FeOOH) [30], specifically akaganeite (β-FeOOH) [31]. In fact, crystals of this β-FeOOH phase typically have rod or needle-like shapes [32]. A closer inspection of the SEM images in Figure 6c (inset) show that the small crystallites of the new phase are agglomerates of needle-shape nanometric crystals.

The formation of iron oxyhydroxides has been observed in the corrosion process of iron in the presence of chloride ions [33]. It has been reported by [34] that when iron metals are exposed to concentrated hydrochloric acid and subsequently to a humid environment, FeCI_2_.4H_2_O crystals are first formed. After some time, a membrane of β-FeOOH covers the ferrous chloride. β-FeOOH is commonly found on iron exposed to highly chlorinated water or on iron immersed in strong solutions of hydrochloric acid or sodium chloride that are then exposed to air [34]. A similar situation could be occurring for our crystals: first, the formation of the FeCl_2_·4H_2_O crystals by the as-described process, an iron compound exposed to concentrated HCl solution, and then exposed to air for few days (with the corresponding air humidity). In addition, iron chloride hydrates are deliquescent. Thus, ferrous chloride could absorb air humidity and dissolve in it, leading to a superficial film solution formed by Fe^2+^, Cl^−^, and oxygen, promoting the oxidation of Fe(II) to Fe(III), forming the β-FeOOH phase.

For the AAFC samples (aged for a period of 2 months), a shaper peak at 331 cm^−1^ can also be detected in some areas (Figure 9b), which can be also related to the akaganeite phase with Cl [35]. All of these results point toward the formation of akaganeite iron oxyhydroxide due to the presence of HCl on the surface of the crystals that remain after the washing process.

Photoluminescence (PL) measurements were also performed on both green and orange regions. Again, a clear difference recorded signal was observed (Figure 10). The PL spectrum of green crystals is dominated by a sharp peak at 365 nm. No reference in the literature has been found about the PL emission of the FeCl_2_·4H_2_O compound, but the same emission is found in the CFC and FFC samples (Figure 10, blue and green lines, respectively). Due to the sharpness and shape of this peak, it can be related to the near band edge emission of the material. This emission disappears completely in the orange regions of the AFC sample. As it has been already described, the orange regions are composed of the β-FeOOH phase, so the luminescence emission related to FeCl_2_·H_2_O is absent. (Figure 10, orange line).

### 3.4. Mössbauer Spectroscopy

Figure 11 collects the 298 K ^57^Fe Mössbauer spectra recorded from a commercial FeCl_2_·4H_2_O sample (CFC), a fresh sample (FFC), and an aged sample (AFC). The CFC spectrum consists of a narrow doublet with isomer shift δ = 1.21 mm·s^−1^ and quadrupole splitting δ = 2.98 mm·s^−1^. These hyperfine parameters are fully characteristic of FeCl_2_·4H_2_O [36]. The spectrum taken from FFC is also a narrow doublet with identical parameters to those of CFC. Therefore, we can confidently ensure that the fresh sample FFC is FeCl_2_·4H_2_O. In contrast, to the spectra of these two samples, the spectrum recorded from the aged AFC sample shows two quadrupole doublets. A major one, accounting for 95% of the spectral area and parameters typical of FeCl_2_·4H_2_O, and a minor one, 5%, with isomer shift δ = 0.38 mms^−1^ and quadrupole splitting Δ = 0.54 mms^−1^. These Mössbauer parameters can be assigned to Fe^3+^ oxyhydroxides such as lepidocrocite (γ-FeOOH) or akaganeite (β-FeOOH) [37]. Since akageneite is usually formed in the presence of chloride ions, it is plausible to think that the Fe^3+^ doublet could be due to this latter oxyhydroxide. Additionally, akaganeite is the compound detected in Raman spectroscopy, as we have described before. The spectra of samples aged for longer (AAFC, 2 months) did not show higher Fe^3+^ concentrations than those shown by AFC.

The observed colour change can then be related to a degradation process of the crystals due to the formation of rust on the surface and the intergranular sites (Figure 2d). The formed rust is mainly composed by akaganeite iron oxyhydroxide (β-FeOOH), as confirmed by the Raman and Mössbauer measurements. In the case of the sample aged for longer, the formation of a surface film of the akaganeite phase (which is not liquescent) could reduce the degradation of the ferrous crystals. For this reason, the corrosion stops when this film is thick enough to prevent the contact of the FeCl_2_·4H_2_O surface with the air humidity.

In addition, it was possible to obtain bigger FeCl_2_·4H_2_O from the water pickling liquors by means of the slowly concentrated ferrous phase (Figure 12a). The Raman spectrum of these crystals is the same as those described in Figure 5, confirming that they have the same crystal structure (Figure 12b). More interesting is that these bigger crystals do not present degradation even if they are left in ambient conditions for a long period of time. As it can be seen in the crystal on the right of Figure 12a, yellow to orange regions are sometimes visible (check the right part of the crystals). These agglomerates already appear in the FFC big crystals, and they do not change in size or color over time. The Raman spectrum show that they are related to the FeCl_3_ compound without modes related to the oxyhydroxides. This indicates that the corrosion process observed in the smaller crystals is not produced in the bigger ones.

## 4. Conclusions

Ferrous chloride crystals with FeCl_2_·4H_2_O stoichiometry were obtained in a pilot evaporation-crystallization plant using pickling water from the company HIASA (Asturias, Spain). The effect of aging of the samples on their structural characterization was also assessed. XRD and Rietveld refinement in both the fresh and aged samples attributed to the FeCl2·4H2O structure previously reported for this type of structure. SEM images revealed a smooth surface that could be observed in the case of the fresh sample, while in the case of the aged sample, some rougher surface areas were found. µ-Raman spectra for the reference sample and the obtained ferrous chloride crystals were similar, and this can be attributed to FeCl_2_·4H_2_O phase. A total of two different Raman spectra were found in the case of the aged sample. Green regions exhibit the same Raman spectra attributable to the FeCl_2_·4H_2_O phase. In the case of the orange regions, Raman peaks that can be related to different phases of iron were found probably due to a corrosion process of iron in the presence of chloride ions in the aged sample. Differences in the PL intensity were observed for the analyzed samples. Despite that, to the best of our knowledge PL study of this type of compound has not previously been reported, and the dominated sharp band could be related to the near band edge emission of the material. Mössbauer spectra of both the reference sample and the obtained crystals consist of a narrow doublet, which is fully characteristic of FeCl_2_·4H_2_O. In the case of the aged sample, the spectrum shows two quadrupole doublets. One of them accounts for 95% of the spectral area, which is typical of FeCl_2_·4H_2_O, while the other one accounts for 5% and can be assigned to Fe^3+^ oxyhydroxides. The spectra of the samples aged for longer periods did not show higher Fe^3+^ concentrations. Thus, all of the obtained results reveal that it is possible to obtain high-purity and high-stability FeCl_2_·4H_2_O crystals from water pickling liquors.

## Figures and Tables

**Figure 1 materials-14-04840-f001:**
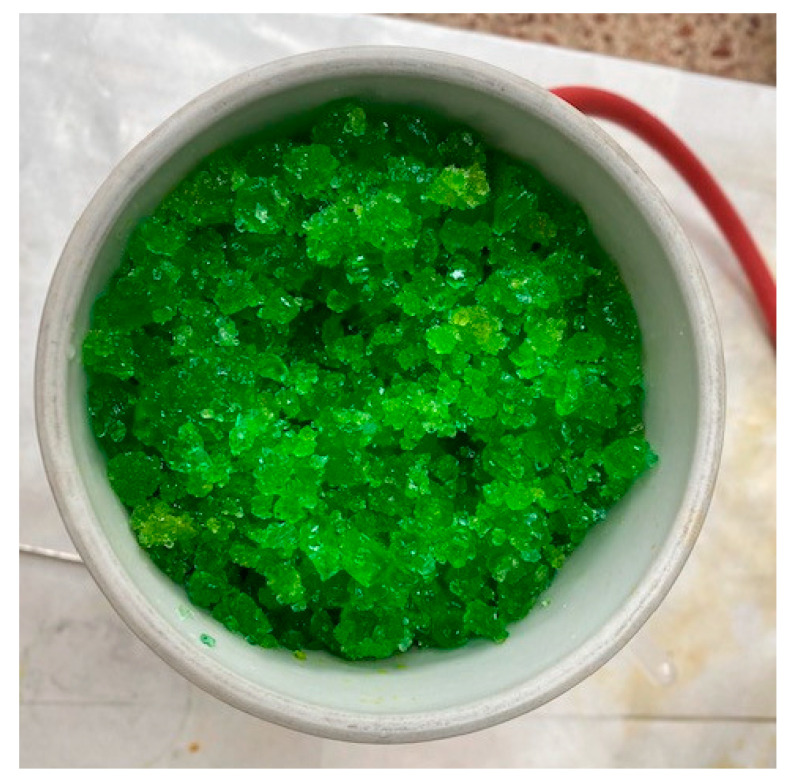
Ferrous chloride crystals obtained from water pickling liquors by the described method.

**Figure 2 materials-14-04840-f002:**
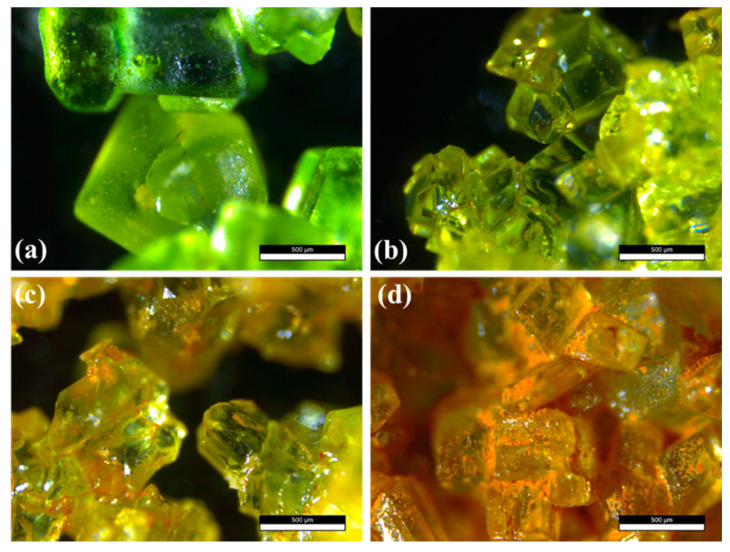
Optical micrographs of the FeCl_2_·4H_2_O crystals: (**a**) commercial reference crystals (CFC). (**b**) Fresh FeCl_2_·4H_2_O sample (FFC). (**c**,**d**) Images showing different stages of the aging of the crystals: (**c**) AFC; (**d**) AAFC. In all cases, the scale bar is 500 μm.

**Figure 3 materials-14-04840-f003:**
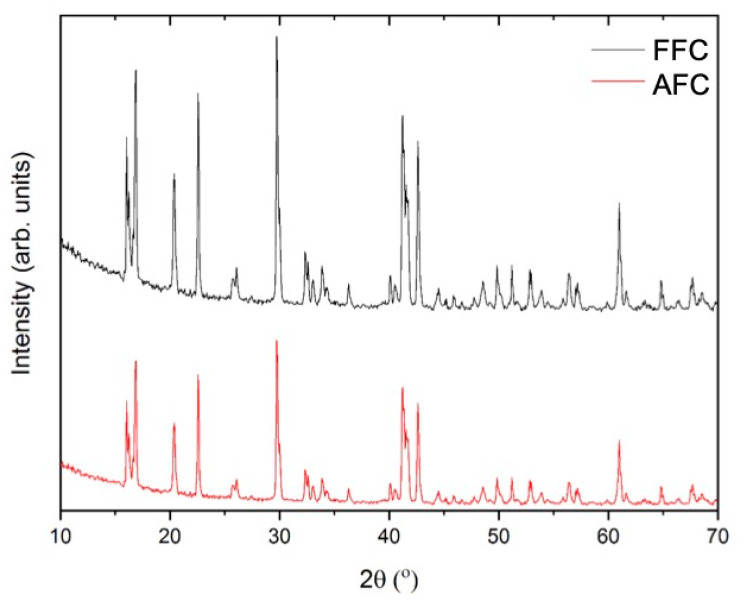
XRD patterns of the obtained ferrous chloride samples.

**Figure 4 materials-14-04840-f004:**
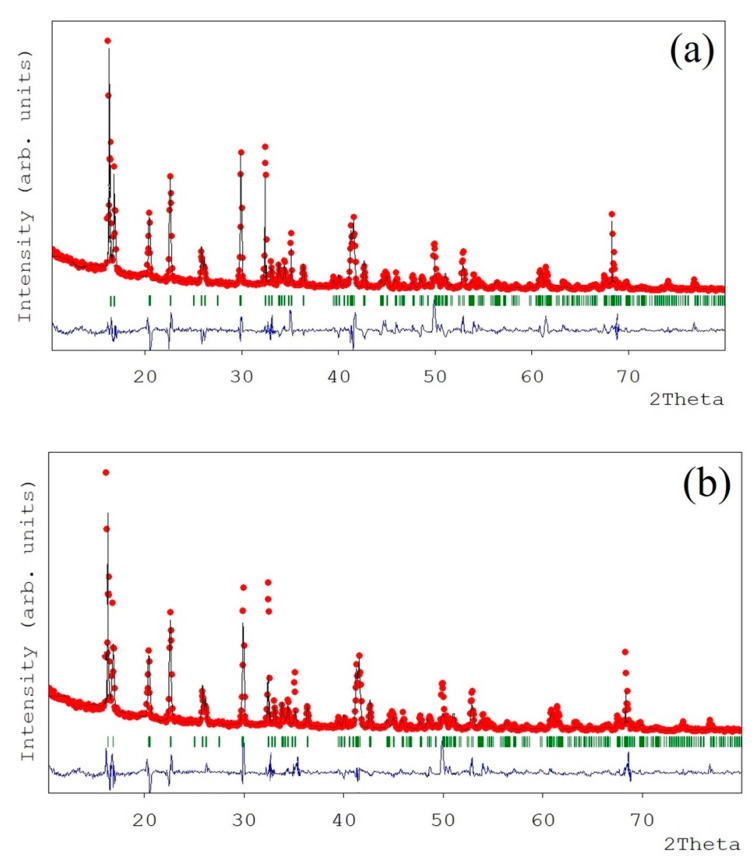
Observed, calculated, and difference XRD patterns obtained in the Rietveld refinements for (**a**) FFC and (**b**) AFC samples.

**Figure 5 materials-14-04840-f005:**
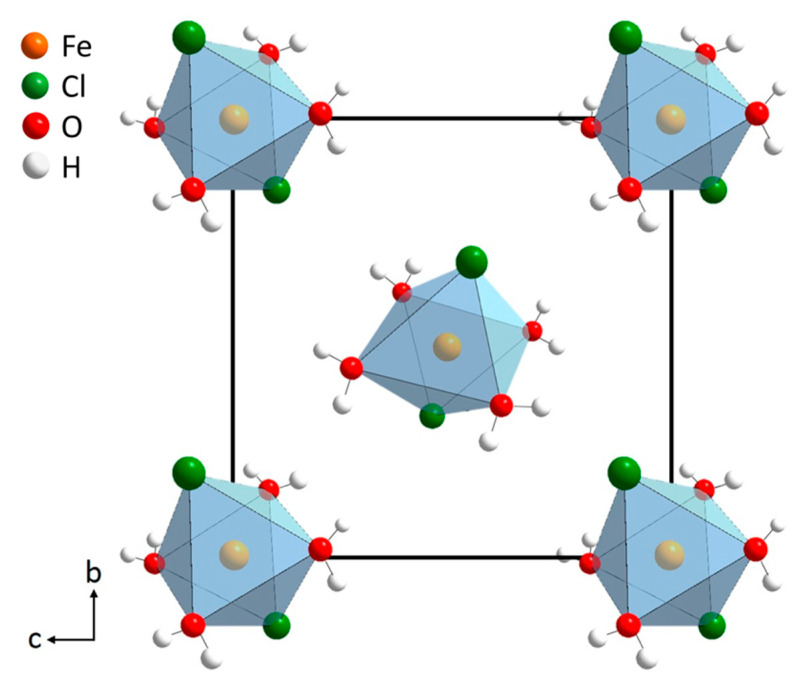
Crystal structure for the FeCl_2_·4H_2_O using Diamond 3.0 software.

**Figure 6 materials-14-04840-f006:**
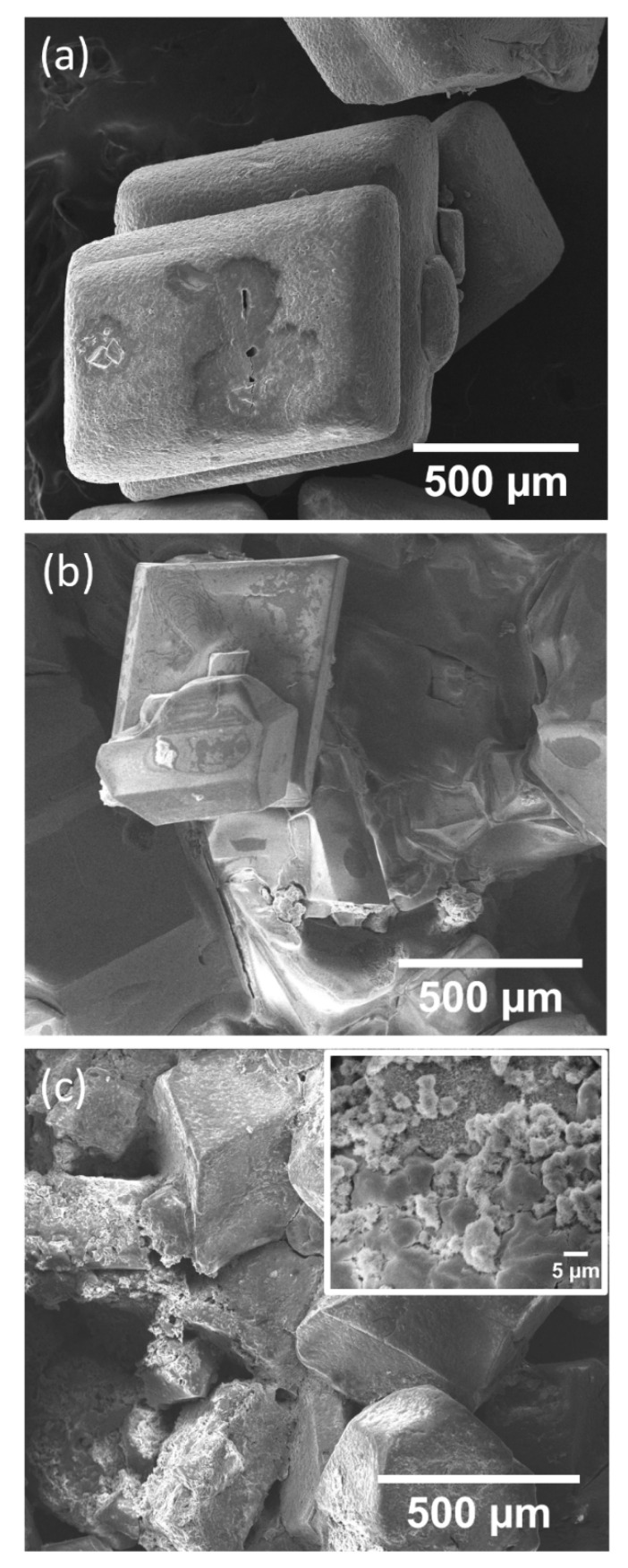
SEM micrographs of the (**a**) CFC reference sample and the (**b**) FFC and (**c**) AFC obtained crystals.

**Figure 7 materials-14-04840-f007:**
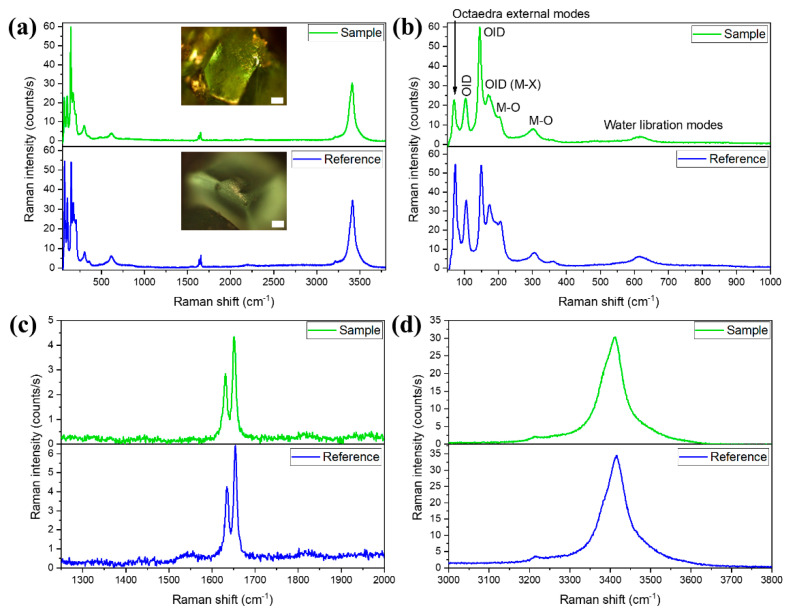
Micro-Raman spectra recorded on the CFC sample (blue) and the FFC sample (green). The excitation wavelength is 633 nm. (**a**) Spectra showing all of the Raman peaks obtained. In the inset, optical images of both reference and obtained crystals are shown. (**b**) Raman modes appearing in the low-frequency range (50–1000 cm^−1^). (**c**) Intermediate frequency range (1250–2000 cm^−1^). (**d**) High-frequency range (3000–3800 cm^−1^). In all cases, the scale bar is 100 μm.

**Figure 8 materials-14-04840-f008:**
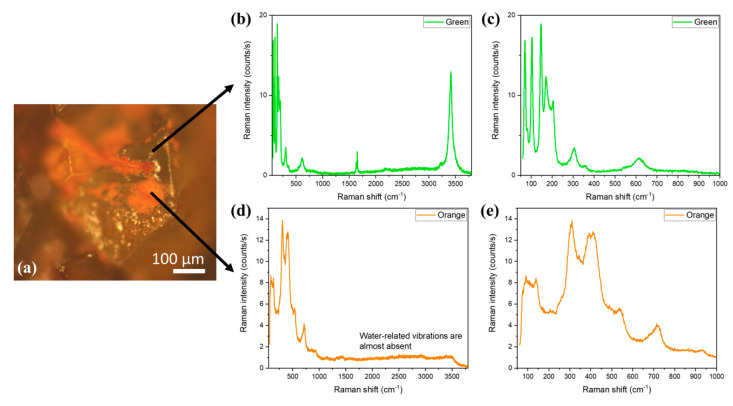
Micro-Raman spectra recorded on AFC sample. (**a**) Optical micrograph of a crystal, showing the regions where the spectra were taken. (**b**,**c**) Raman spectra measured on the green region. (**d**,**e**) Raman spectra recorded on the orange region.

**Figure 9 materials-14-04840-f009:**
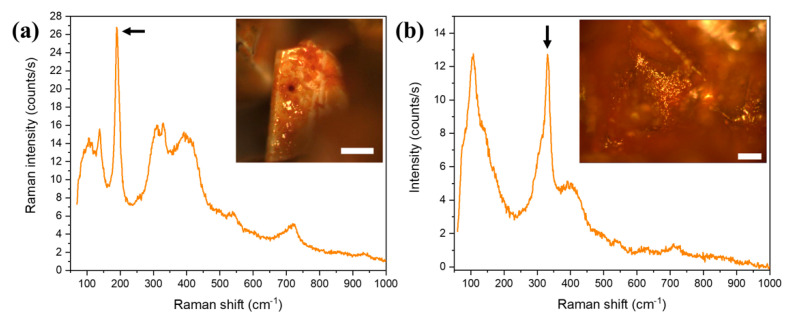
Micro-Raman spectra recorded on AAFC sample: (**a**) showing the peak related to FeCl_3_; (**b**) showing the peak adscribed to akaganeite phase with Cl. Insets: optical micrographs of the crystals where the spectra were taken. In all cases, the scale bar is 100 μm.

**Figure 10 materials-14-04840-f010:**
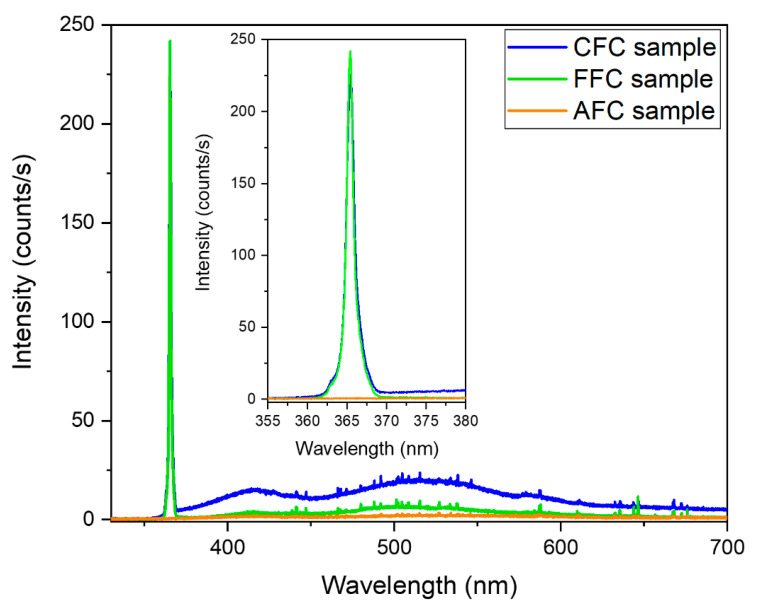
PL emission of CFC sample, FFC sample, and AFC sample (orange region) recorded using a 325 nm laser excitation wavelength. In the inset, a zoom of the peak at 365 nm is shown.

**Figure 11 materials-14-04840-f011:**
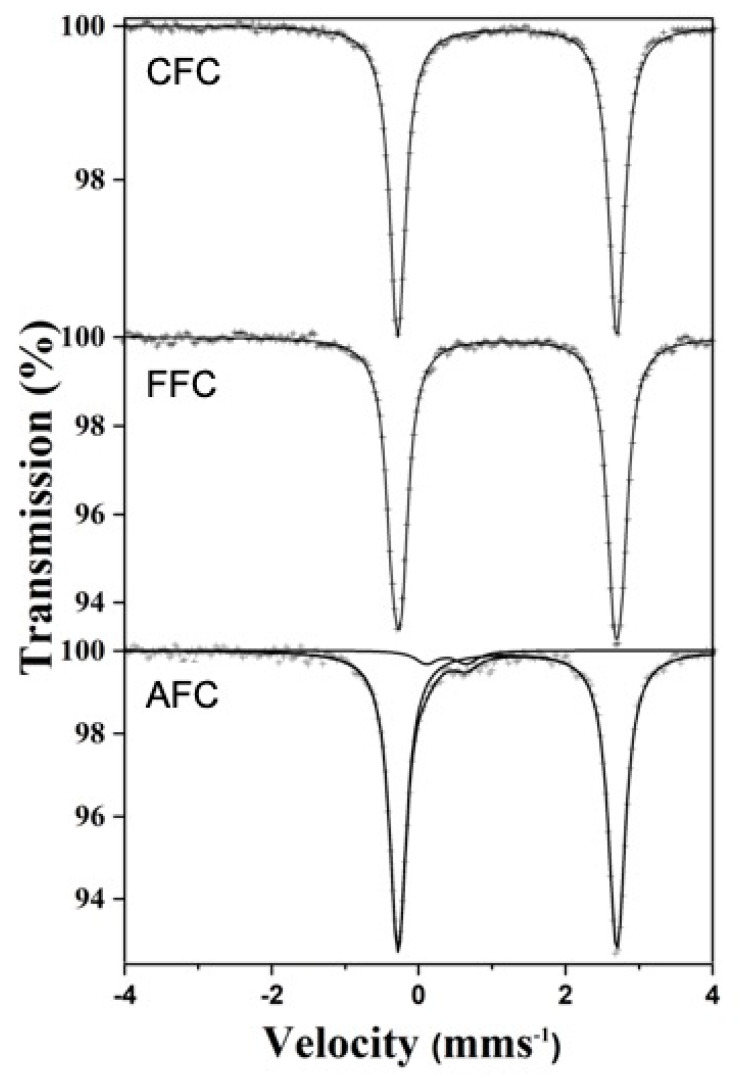
298 K ^57^Fe transmission Mössbauer spectra recorded from the various samples.

**Figure 12 materials-14-04840-f012:**
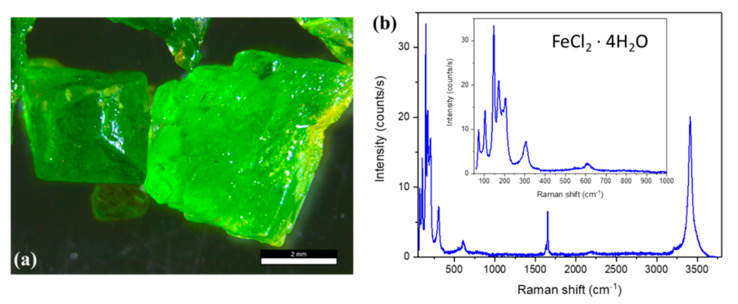
(**a**) Optical micrographs of the FeCl_2_·4H_2_O crystals of larger size (scale bar of 2 mm). (**b**) Micro-Raman spectra recorded on the crystals of part (**a**).

**Table 1 materials-14-04840-t001:** Nomenclature and aging description of the samples.

Sample	Name	Description
Commercial ferrous chloride	CFC	FeCl_2_·4H_2_O crystals purchased from Merck ≥99.0%
Fresh ferrous chloride	FFC	Crystals obtained from water pickling liquors
Aged ferrous chloride	AFC	Crystals aged for 10 to 20 days
Aged ferrous chloride	AAFC	Crystals aged for 2 months

**Table 2 materials-14-04840-t002:** Lattice parameters and cell angles calculated from the Rietveld refinements for the obtained samples.

Sample	a (Å)	b (Å)	c (Å)	α (°)	β (°)	γ (°)		R_wp_	R_F_	R_B_
[26]	5.885(3)	7.174(3)	8.505(4)	90	111.11(5)	90	--	--	--	--
FFC	5.891(4)	7.117(4)	8.510(1)	90	111.10(2)	90	17.1	18.3	5.58	4.18
AFC	5.902(8)	7.121(6)	8.486(3)	90	112.02(8)	90	16.6	19.4	6.10	3.83

## Data Availability

Data sharing is not applicable to this article.

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
