# Peer review of "Obtention and Characterization of Ferrous Chloride FeCl2·4H2O from Water Pickling Liquors"

_materials, 2021, doi:10.3390/ma14174840_

Round 1

Reviewer 1 Report

The authors present the transformations that take place over time for FeCl2.4H2O comparing CFC, CFF and CFT which are analyzed by different methods: XRD, Micro-Raman, Photoluminescence, Optical Micrograph, Field emission scanning electron microscopy and Mosbauer spectroscopy. The conclusions should be shorter and clearer. If the authors have the opportunity they could do (although there are already many methods) TG to see when the water of crystallization is lost. 

Author Response

We would like to thank the reviewer for his/her interesting and useful comments. In the attached file, we answer his/her questions on a point by point basis and indicate how the text has been modified.

Reviewer 2 Report

The manuscript entitled “Obtention and characterization of ferrous chloride FeCl2·4H2O from pickling waste liquors” presents some information regarding the precipitation and instability of iron chloride hydrates from highly concentrated solutions from pickling.

The manuscript presented has, in my opinion, several lacks which make it hard to understand and read, and this is why I have to consider it as not ready to be published (reject). However, it can be improved/changed.

Authors follow a recipe to get crystals from the pickling solution in order to obtain a method to recover iron from solutions, and somehow neutralize the hazardous of this potential concern. Then, authors compare those crystals with a reference (commercial crystals with the same composition and structure). Regrettably, the precipitated crystals are not stable and react with the moisture triggering its transformation into other iron phases.

So, first question is: why is it important for the reader (there is already a commercial process which obtains a stable iron phase)?? It can be deduced along the text that authors already knew that the corrosion process would happen. It would be good to compare with other paths in order to stabilize the desired phase and stop the corrosion process.

Second, in my opinion, for experimental works, timing is basic. Along the text, authors do not pay much attention on this, making the reader get lost.

Third, maybe there is too much information with no additional improvements. I mean, XRD (must improve), SEM (should improve) and Raman (must assign all vibrations/bands) should be enough to explain what authors are showing here. PL and Mössbauer techniques, although they are of course interesting, maybe need to be better explained (why it is important to include) or it makes the reader get lost with data.

English writing need to be improved.

Figures, at least SEM images, need to be improved. Undoubtedly, it will enhance the quality of the manuscript (and let the authors make a truly reference to the observations conducted).

Attached you can find the reviewed manuscript with some other concerns.

Author Response

(The authors gave the same response as above.)

Round 2

Reviewer 1 Report

Obtaining FeCl2 · 4H2O crystals from residues is an interesting topic especially in terms of industrial applications. FeCl2 · 4H2O crystals were investigated by several methods: X-ray diffraction, scanning electron microscopy Mössbauer spectroscopy, photoluminescence. The manuscript has been improved and the presentation is clearer and I propose that it be accepted.